# Silencing the Mitochondrial Gatekeeper VDAC1 as a Potential Treatment for Bladder Cancer

**DOI:** 10.3390/cells13070627

**Published:** 2024-04-04

**Authors:** Belal Alhozeel, Swaroop Kumar Pandey, Anna Shteinfer-Kuzmine, Manikandan Santhanam, Varda Shoshan-Barmatz

**Affiliations:** 1Department of Life Sciences, Ben-Gurion University of the Negev, Beer-Sheva 84105, Israel; belala@post.bgu.ac.il (B.A.); pandey.swaroop@gmail.com (S.K.P.); santhana@post.bgu.ac.il (M.S.); 2National Institute for Biotechnology in the Negev, Ben-Gurion University of the Negev, Beer-Sheva 84105, Israel; shteinfe@post.bgu.ac.il

**Keywords:** bladder cancer, mitochondria, si-RNA, VDAC1

## Abstract

The strategy for treating bladder cancer (BC) depends on whether there is muscle invasion or not, with the latter mostly treated with intravesical therapy, such as with bacillus Calmette–Guérin (BCG). However, BCG treatment is unsuccessful in 70% of patients, who are then subjected to radical cystectomy. Although immune-checkpoint inhibitors have been approved as a second-line therapy for a subset of BC patients, these have failed to meet primary endpoints in clinical trials. Thus, it is crucial to find a new treatment. The mitochondrial gatekeeper protein, the voltage-dependent anion channel 1 (VDAC1), mediates metabolic crosstalk between the mitochondria and cytosol and is involved in apoptosis. It is overexpressed in many cancer types, as shown here for BC, pointing to its significance in high-energy-demanding cancer cells. The BC cell lines UM-UC3 and HTB-5 express high VDAC1 levels compared to other cancer cell lines. VDAC1 silencing in these cells using siRNA that recognizes both human and mouse VDAC1 (si-m/hVDAC1-B) reduces cell viability, mitochondria membrane potential, and cellular ATP levels. Here, we used two BC mouse models: subcutaneous UM-UC3 cells and chemically induced BC using the carcinogen *N*-Butyl-*N*-(4-hydroxybutyl) nitrosamine (BBN). Subcutaneous UM-UC3-derived tumors treated with si-m/hVDAC1 showed inhibited tumor growth and reprogrammed metabolism, as reflected in the reduced expression of metabolism-related proteins, including Glut1, hexokinase, citrate synthase, complex-IV, and ATP synthase, suggesting reduced metabolic activity. Furthermore, si-m/hVDAC1-B reduced the expression levels of cancer-stem-cell-related proteins (cytokeratin-14, ALDH1a), modifying the tumor microenvironment, including decreased angiogenesis, extracellular matrix, tumor-associated macrophages, and inhibited epithelial–mesenchymal transition. The BBN-induced BC mouse model showed a clear carcinoma, with damaged bladder morphology and muscle-invasive tumors. Treatment with si-m/hVDAC1-B encapsulated in PLGA-PEI nanoparticles that were administered intravesically directly to the bladder showed a decreased tumor area and less bladder morphology destruction and muscle invasion. Overall, the obtained results point to the potential of si-m/hVDAC1-B as a possible therapeutic tool for treating bladder cancer.

## 1. Introduction

Bladder cancer (BC) is the tenth most commonly diagnosed malignancy, with a global incidence of more than the 400,000 new cases reported annually and nearly 200,000 deaths [1]. It is three times more common in men, and fatality is inevitable once it metastasizes [2]. BC is the most common neoplasm of the urinary system. Urothelial carcinoma (UC), also known as transitional cell carcinoma, is the most common histologic type of this cancer (approximately 90%), and other types include squamous cell carcinoma, small-cell carcinoma, and adenocarcinoma [3]. UC can be defined as the invasion of the basement membrane or lamina propria or deeper by neoplastic cells of urothelial origin [3].

There are multiple known risk factors for BC, with tobacco smoking being the main one. In smokers, the risk is two- to six-fold higher than that of non-smokers, with the risk depending on the duration and intensity of smoking. Other risk factors are schistosomiasis infection and occupational exposure to certain chemicals [4,5,6]. BC originates from the urothelium, and most BC is urothelial.

The treatment strategy of UC depends on whether there is muscle invasion. Management of patients with non-muscle-invasive UC includes endoscopic resection and risk-based intravesical therapy, such as bacillus Calmette–Guérin (BCG). BCG has been shown to decrease UC recurrence and progression by up to 37% when compared to chemotherapy [7]. However, this therapy not only works on bladder cancer by activating the immune system but it also induces apoptosis, necrocytosis, and oxidative stress. In addition, BCG induces a high expression of PD-L1, which explains BCG failure and the relapse of BC [8,9]. This treatment is unsuccessful in 70% of patients, who are thus subjected to radical cystectomy, which is a life-changing surgical procedure. Recently, immune-checkpoint inhibitor (ICI) antibodies were initially approved as a second-line therapy for a subset of BC patients. However, the ICIs durvalumab and atezolizumab were withdrawn, given their failure to meet primary endpoints in subsequent clinical trials [10]. Thus, a new treatment is needed.

Growing evidence suggests that metabolism directly supports oncogenic signaling to foster tumor malignancy [11,12]. Thus, targeting cancer cell metabolic alterations may preferentially affect malignant cells and is likely to have broad therapeutic implications.

Voltage-dependent anion channel 1 (VDAC1) is a mitochondrial protein that controls cell energy, metabolic homeostasis, and apoptosis [13]. VDAC1 mediates metabolic crosstalk between the mitochondria and cytosol, transporting metabolites, ions, nucleotides, Ca^2+^, and more that regulate mitochondrial activity. It also plays a key role in apoptosis, participating in the release of apoptotic factors from the mitochondria and interacting with anti-apoptotic regulators [13]. VDAC1 is highly expressed in different tumors [13], pointing to its significance in high-energy-demanding cancer cells. We have demonstrated that abrogation of its expression by 2′-O-Me-modified siRNA specific to human (h)VDAC1 (si-hVDAC1) reduced cellular ATP levels and cell proliferation in a panel of cell lines regardless of cell origin and mutation status. In addition, it inhibits solid tumor development and growth in cervical, lung, and breast cancers and in glioblastoma [14,15,16,17].

si-VDAC1 treatment of several tumor types induces metabolic rewiring of the malignant phenotype of cancer metabolism, resulting in a reversal of oncogenic properties that include reduced tumor growth, invasivity, stemness, epithelial–mesenchymal transition (EMT), and angiogenesis [16]. The reprogrammed cancer cell metabolism alters tumor microenvironment, reducing angiogenesis and altering the expression of the extra cellular matrix and structure-related genes, including collagens and glycoproteins [17]. It also reduces deposition and the number of degradation-related genes [17].

Moreover, VDAC1 depletion not only leads to the reprogramming of cancer cell metabolism, but it also alters several epigenetic-related enzymes and factors such as the acetylation-related enzymes SIRT6, HDAC2, and SIRT1 and changes the acetylation and methylation profiles of histone 3 and histone 4 [18], pointing to an interplay between metabolism and epigenetics. These epigenetic modifications can explain the altered expression levels of about 4000 genes associated with cancer cells, reversing their oncogenic properties [15,17]. As a key regulator of metabolic and energy reprogramming, disrupting cancer energy and metabolism homeostasis by targeting VDAC1 offers a potential anti-cancer therapy strategy [19].

Here, we demonstrate that silenced VDAC1 expression in BC using two BC mouse models, subcutaneous UM-UC3 cells and chemically induced BC using *N*-Butyl-*N*-(4-hydroxybutyl) nitrosamine (BBN), resulted in metabolism reprogramming and led to inhibited tumor growth, as well as modulations of the tumor microenvironment (TME), including reduced angiogenesis, extracellular matrix (ECM), tumor-associated microphages, inhibited stemness, and epithelial mesenchymal transition (EMT). These results indicate the potential of si-VDAC1 as a possible strategy for treating bladder cancer.

## 2. Materials and Methods

### 2.1. Materials

Hematoxylin, eosin, 4′,6-diamidino-2-phenylindole (DAPI), Poly(d,l-lactide-*co*-glycolide) (PLGA, (lactide:glycolide 50:50)), polyethylenimine (PEI, branched, 25 kDa), polyvinyl alcohol (PVA), Triton X-100, and Tween-20 were obtained from Sigma (St. Louis, MO, USA). Paraformaldehyde and formaldehyde were obtained from Emsdiasum (Hatfield, PA, USA). Phosphate-buffered saline (PBS), Dulbecco’s modified Eagle’s medium (DMEM) media, normal goat serum (NGS), the supplement fetal bovine serum (FBS), and penicillin–streptomycin were obtained from Gibco (Grand Island, NY, USA). 3,3-diaminobenzidine (DAB) was obtained from (Vector Laboratories ImmPact-DAB, Burlingame, CA, USA). Alexa-fluor 555-Phalloidin was obtained from Thermo Fisher Scientific (Waltham, MA, USA). The source and the dilutions used of primary and secondary antibodies are presented in Appendix A. A CellTiter-Glo assay kit was obtained from Promega (Madison, WI, USA). Dimethyl sulfoxide (DMSO) was purchased from MP Biomedicals (Solon, OH, USA). In vivo Jet-PEI and the polyvalent Jet Prime transfection reagents were from Polyplus (llkirch-Graffenstaden, France). Matrigel matrix was purchased from Corning (Corning, NY, USA).

### 2.2. Cell Culture

Human-derived UM-UC3, an invasive bladder cancer and HTB-5, a grade IV bone metastatic bladder cancer, both derived from the epithelium; THP-1, human acute monocytic leukemia; A549, human lung adenocarcinoma epithelial cells; U-87MG, human glioblastoma; SH-SY5Y, human neuroblastoma; HEK-293, human embryonic fibroblasts cells; MEFs, mouse embryonic fibroblast cells; and HaCat, immortalized human keratinocyte cell lines were obtained from the American Type Culture Collection (ATCC) (Manassas, VA, USA) and were maintained as per the ATCC instructions. Cells were maintained in ATCC-recommended medium at 37 °C in an incubator with 5% CO_2_. MEC-1, human B-cell chronic lymphocytic leukemia cells were obtained from the Leibniz Institute DSMZ—German collection of microorganisms and cell cultures (Braunschweig, Germany). Cell lines were routinely tested for mycoplasma contamination.

### 2.3. siRNA Transfection

Human and mouse VDAC1-specific siRNA (si-m/hVDAC1-B) and non-targeted siRNA (si-NT) were synthesized and obtained from Gene Pharma (Suzhou, China). The underlined nucleotides are 2′-*O*-methyl-modified. si-m/h-VDAC1-B; Sense: 5′GAAUAGCAGCCAAGUAUCAGtt3′; Anti-sense: 5′CUGAUACUUGGCUGCUAUUCtt3′. si-NT; Sense: 5′GCAAACAUCCCAGAGGUAU3′; Anti-sense: 5′AUACCUCUGGGAUGUUUGC3′.

UM-UC3 and HTB-5 cells were seeded in 6-well plates (3 × 10^5^ cells/well) and were transfected at 40–60% confluence with the indicated concentration of si-m/hVDAC1-B or si-NT using the JeT Prime transfection reagent (Polyplus, llkirch-Graffenstaden, France) according to the manufacturer’s instructions.

### 2.4. Cell Viability Assay

Cells were transfected with si-NT or si-m/hVDAC1, and at 24 h post-transfection, were counted and seeded in 96-well plates. Then, cells were subjected to the XTT assay according to the manufacturer’s instructions (Biological Industries; Beit Haemek, Israel). Absorbance at 450–500 nm, and 630–690 as a reference, were measured using an Infinite M1000 plate reader (Tecan, Männedorf, Switzerland).

### 2.5. Protein Extraction from Cells and Tumors, Gel Electrophoresis, and Immunoblot

Cells were lysed in lysis buffer (50 mM Tris-HCl, pH 7.5, 150 mM NaCl, 1 mM EDTA, 1.5 mM MgCl_2_, 10% glycerol, 1% Triton-X100) supplemented with a protease inhibitor cocktail (Calbiochem, San Diego, CA, USA) and incubated on ice (30 min). Then, cell lysate was centrifuged (10 min, 15,000× *g* at 4 °C).

Tissues were lysed in lysis buffer [4% sodium dodecyl sulfate (SDS), 100 mM Tris-HCl, pH = 8.0, 5 mM dithiothreitol (DTT)] freshly supplemented with 0.5 mM ethylene glycol-bis(β-aminoethyl ether)-*N*,*N*,*N*′,*N*′-tetraacetic acid (EGTA) and 0.5 mM phenylmethylsulfonyl fluoride (PMSF). Then, tissues were homogenized and incubated for 3 min at 95 °C, sonicated, and centrifuged (10 min, 15,000× *g*, RT). In both cases, the supernatant was subjected to protein determination using a Lowry assay and stored in −80 °C until analyzed by gel electrophoresis and immunoblotting.

Protein samples (10–20 μg) were resolved by SDS polyacrylamide gel electrophoresis and immunoblotted using the selected primary antibodies (Appendix A), followed by incubation with horseradish peroxidase (HRP)-conjugated secondary antibodies. HRP activity was determined using enhanced chemiluminescent substrate (Advantsa, San Jose, CA, USA). Band intensity was quantified using Image J software (version 1.8.0) or FUSION-FX (Vilber Lourmat, France).

### 2.6. Mitochondrial Membrane Potential Determination and Cellular ATP Levels

Mitochondrial membrane potential was determined using tetramethylrhodamine (TMRM), a potentially sensitive dye. Cells were transfected with si-NT or si-m/hVDAC1-B and, 48 h post-transfection, were seeded in 96 white plates, incubated with TMRM (800 nM, 20 min, 37 °C, 5% CO_2_). The cells were then washed twice with PBS and incubated with media (contains 1% FBS) without phenol red. TMRM fluorescence was measured with an Infinite M1000 plate reader (Tecan, Männedorf, Switzerland).

Cellular ATP levels were estimated using a luciferase-based assay (CellTiter-Glo, Promega) according to the manufacturer’s protocol, and luminescence was recorded using an Infinite M1000 plate reader (Tecan, Männedorf, Switzerland).

### 2.7. Immunocytochemistry (IF)

Cells were seeded on sterile glass coverslips in 12-well plates and cultured until reaching about 80% confluence. Cells were transfected with si-NT or si-m/hVDAC1-B and 72 h post-transfection, washed with PBS, fixed in 4% paraformaldehyde (20 min), washed three times with PBS, permeabilized with 0.3% Triton X-100 in PBS, and blocked in blocking buffer (10% NGS, 1% fatty-acid-free bovine serum albumin [BSA], 0.1% Triton X100 diluted in PBS) for 2 h. Cells were incubated overnight at 4 °C with the appropriate primary antibodies (Appendix A), followed by three washes with PBS. Then, they were incubated with fluorescent-conjugated secondary antibody (Appendix A) for 2 h at room temperature (RT) in the dark. Following a wash with PBS, cells were incubated with DAPI (0.5 μg/mL) for 15 min in the dark and carefully washed, dried, and mounted on slides with fluoroshield mounting medium (Immunobioscience, Mukilteo, WA, USA). After overnight drying at 4 °C, images were acquired with a confocal microscope (1X81; Olympus, Tokyo, Japan).

### 2.8. Preparation of siRNA Loaded–PLGA-PEI Nanoparticles

The PLGA-PEI-PVA-siRNA-loaded complexes were prepared by a solvent displacement method with some modifications, as previously reported by [20,21]. 

To the PEI-aqueous solution (10 mg/mL), PVA solution was added (1 mg/mL) and mixed well by vortexing. si-NT or si-m/hVDAC1-B solutions in nuclease-free water were added to the PEI-PVA solution and mixed well by vortexing. Samples were incubated in a 37 °C water bath for 30 min. After the incubation, the PEI-PVA-siRNA mixture was added to a PLGA-acetone solution (10 mg/mL) by pipetting.

Primary emulsion, PLGA-PEI-siRNA, was added in a drop-wise manner (0.5 mL/min) to the glass beaker containing nuclease-free water with PVA solution. The mixture was continuously stirred with a magnetic stirrer at RT in a chemical hood for 2–3 h until complete evaporation of the organic solvent (no acetone smell) occurred. The milky mixture was centrifuged at 20,000× *g* at 4 °C for 30 min. The supernatant was removed, and the pellet was re-suspended in 10 mL nuclease-free water and centrifuged at 20,000× *g* at 4 °C for 30 min. The pellet was re-suspended in PBS in nuclease-free water and stored at −20 °C until use.

### 2.9. Bladder Cancer Mouse Models

Female athymic nude mice (6–8 weeks old) were procured from Envigo (Indianapolis, IN, USA). Mice were housed (4 animals per cage) with a 12/12 h light/dark cycle and with ad libitum access to food and water. All experimental protocols were approved by the Institutional Animal Care and Use Committee. 

Mouse xenograft model—Human bladder cancer UM-UC3 cells (1.2 × 10^6^ cells) were implanted subcutaneously with a Matrigel matrix on the dorsal flanks. Tumor growth was recorded using digital calipers, and tumor volume was calculated using the formula (*π*/6) * (*L* × *W*2) (*L* = length; *W* = width). Mice were randomly divided into three groups once the average tumor volume reached ~50 mm^3^ and were treated three times a week intratumorally with si-NT (50 nM) or si-m/hVDAC1-B (50 nM and 200 nM) using in vivo JetPEI reagent (Polyplus, llkirch-Graffenstaden, France). Tumors were intratumorally treated until the injection volume was 10–15% of the tumor volume. To ensure that the siRNAs could reach all of the tumor volumes, the tumor was injected at a single point (bolus) if the tumor was small, and at up to three different boluses for a large tumor. In addition, the tumor injection points were not constant; they were rotated with every injection. When the tumor size reached 1000 mm^3^, mice were sacrificed using CO_2_ gas, and the tumors were excised, photographed, and weighed. Half of each tumor was fixed and processed for immunohistochemistry (IHC) or immunofluorescence (IF), with the other half frozen in liquid nitrogen for immunoblot.

BBN-induced bladder cancer—BBN-induced bladder cancer was carried out as described previously [22]. Six- to eight-week-old C57BL/6JOlaHsd 10 mice (Envigo, Indianapolis, IN, USA) were treated with BBN (0.05%) in drinking water for a period of 16 weeks. Mice were monitored with ultrasound (VEVO 3100, Visual Sonic, Toronto, ON, Canada) for tumor development and intravesical treatment. The treatment was administered twice a week for a duration of four weeks, starting from week 16 post-BBN treatment, using ultrasound to monitor the injection. Five mice were treated with si-NT (240 nM), and another five were treated with si-m/hVDAC1-B (240 nM) encapsulated in PLGA-PEI nano-particles, prepared as described previously [16] with some modifications. Ultrasound images were also used to monitor the bladder morphology and tumors. At the end of the experiment (at week 20), mice were sacrificed, bladders were dissected, fixed with 4% formalin, and processed for hematoxylin–eosin (H&E), IHC, or IF staining. To represent the whole tumor area, images were taken from different fields of the tumor section. 

### 2.10. Histological, Immunohistochemistry, and Immunofluorescence Analyses of Bladder Cancer

Formalin-fixed and paraffin-embedded 5 μm thick tumor tissue sections were deparaffinized by heating the slides on a 60 °C hot plate for 1 h with xylene. Thereafter, the sections were rehydrated using a graded ethanol series (100–50%) and subjected to antigen retrieval using either 0.01 M citrate buffer (pH 6.0) or 0.01 M Tris-EDTA (pH 9) at 95–98 °C for 30 min. The sections were incubated in blocking buffer (10% NGS, 1% BSA, and 0.1% Triton) for 2 h and then incubated overnight at 4 °C with primary antibodies (Appendix A) in an antibody buffer (5% NGS, 1% BSA). 

For immunohistochemical staining, endogenous peroxidase activity was blocked by incubating the sections in 3% H_2_O_2_ for 15 min. After washing with (PBS containing 0.01% Triton) PBST, the sections were incubated for 2 h with the appropriate secondary HRP-conjugated antibodies. Sections were washed in PBST, and peroxidase activity was visualized by incubating the section with 3,3-diaminobenzidine (DAB) (ImmPact-DAB, Burlingame, CA, USA). After rinsing in water, the sections were counterstained with hematoxylin and mounted with EUKITT mounting medium (Orsatech, London, UK). Finally, the sections were imaged under a panoramic scanner (3DHISTECH Ltd., Budapest, Hungary) with the same light intensity and exposure time. Quantification of the immunostained images was carried out using HistoQuant software (Quant Center 2.0 software, 3DHISTECH Ltd., Budapest, Hungary).

For immunofluorescence, after washing with PBST, sections were incubated with fluorescent-tagged secondary antibodies (Appendix A) for 2 h at room temperature in the dark. Following a wash with PBST, sections were incubated with DAPI (0.07 μg/mL) for 15 min in the dark, washed, mounted with Fluoroshield mounting medium (Immunobioscience, Mukilteo, WA, USA), and imaged by confocal microscopy (Olympus 1X81). To represent the whole tumor area, images were taken from different fields of the tumor section. 

Human tissue array slides (BL1002a) were obtained from BIOMAX, containing 40 urothelial and 10 normal bladder tissues, with duplicate cores per case. The tissues were IHC-stained using anti-VDAC1 specific antibodies.

### 2.11. Sirius Red and Hematoxylin and Eosin Staining

Sirius red staining was performed on fixed and paraffin-embedded tumor sections. The sections were first deparaffinized and rehydrated with a graded ethanol series. Hematoxylin was used to stain the nuclei for 8 min, followed by a wash with running tap water. The sections were then incubated with a 0.1% Sirius red–1.3% picric acid solution for 1 h. After that, the sections were rapidly washed with 0.5% acetic acid, dehydrated in three changes of 100% ethanol, and cleared in xylene. Finally, the sections were mounted with EUKITT mounting medium (Orsatech, London, UK). The stained sections were visualized with an Olympus LX2-KSP microscope or a panoramic scanner (3DHISTECH Ltd., Budapest, Hungary) and quantified using HistoQuant software (Quant Center 2.0 software).

Hematoxylin–eosin (H&E) staining was performed as described previously [23].

### 2.12. Statistical Analysis

Data are shown as the means ± SEM obtained from at least three independent experiments, unless specified differently. The significance of the differences was calculated by Student’s *t*-test and is reported as * *p* ≤ 0.05, ** *p* ≤ 0.01; *** *p* ≤ 0.001; **** *p* < 0.0001.

## 3. Results

### 3.1. VDAC1 Is Highly Expressed in Human Bladder Cancer Tissue Compared to Healthy Tissues

The expression levels of VDAC1 in the BC tissue array comprising ten samples from human healthy donors and 40 from BC patients (duplicate cores per case) were analyzed by immunohistochemistry (IHC) using anti-VDAC1 antibodies. Representative images from three healthy bladder tissues and nine BC tissues sub-grouped to low (I), medium (II), and high (III) staining intensity are shown (Figure 1A). The staining intensity was quantified in each sub-group I, II, and III and presented as a percent of BC tissues stained at the indicated intensity (Figure 1B). VDAC1 levels were increased by about 5.5-fold in 16% of the BC tissues, by 3.5 in 58%, and by 1.5-fold in 26% of the tissues compared to the VDAC1 expression levels in healthy tissues (Figure 1B).

VDAC1 expression levels were analyzed in several cell lines in comparison to their levels in MEFs (Figure 1C,D). The highest VDAC1 levels were found in the BC cell line UM-UC3 compared to neuroblastoma SHSY-5Y, lung cancer A549, the leukemia THP-1 and MEC-1 and glioblastoma U-87MG, and the HTB-5 high-graded bladder-derived cell line, while transformed HEK-293, HaCaT, and MEF cell lines showed the lowest levels of VDAC1.

To address the question of whether the increased VDAC1 level was due to the higher mitochondrial content, we analyzed the levels of the mitochondrial enzyme citrate synthase (CS). Similar expression levels of CS were found in all of these cell lines, indicating that the increase in VDAC1 expression is per mitochondria and not due to an increase in the number of mitochondria (Figure 1C,D).

### 3.2. VDAC1 Silencing Reduces Cell Viability, Proliferation, Mitochondrial Membrane Potential, and ATP Levels in Bladder Cancer Cell Lines

Next, we tested the effects of si-m/hVDAC1-B, recognizing both mouse and human VDAC1 [24] on the proliferation and survival of the BC cell lines, UM-UC3 and HTB-5. si-m/hVDAC1-B decreased VDAC1 expression by about 80%, relative to si-NT treated cells (Figure 2A,B). Cells treated with si-m/hVDAC1-B showed decreased cell viability, as analyzed by the XTT assay (Figure 2C), as well as mitochondrial membrane potential and cellular ATP levels (Figure 2D,E). In addition, si-m/hVDAC1-B reduced cell proliferation, as reflected in the decreased expression of the proliferation marker, KI-67, to a higher extent in the UM-UC3 than in the HTB-5 cells, by 70% and 30%, respectively (Figure 2F,G), in agreement with higher VDAC1 levels in the UM-UC3 (Figure 1C,D). The expression levels of cyclin D1, a protein required for G1-S phase transition [25], were reduced in both cell lines by reducing VDAC1 expression levels (Figure 2H,I). Finally, UM-UC3 and HTB-5 cells treated with si-m/hVDAC1-B showed a larger size (Appendix A).

### 3.3. VDAC1 Silencing Reduced Tumor Growth of UM-UC3 Cell-Derived Tumor

Next, the effect of VDAC1 silencing on the growth of UM-UC3 cell subcutaneous (s.c.) tumor xenografts established in nude mice was tested. Following tumor formation, mice were split into three matched groups, injected every 3 days with si-NT (50 nM) or si-m/hVDAC1-B (50 nM or 200 nM), and tumor growth was followed. In si-NT-injected tumors, tumor volume grew exponentially and increased by ~20-fold within 35 days, whereas in si-m/hVDAC1-B-treated tumors (TTs), growth was markedly inhibited by about 85% relative to that of si-NT-TTs (Figure 3A). Tumors were removed and photographed (Figure 3B) and showed about an 80% decrease in tumor weight of si-m/hVDAC1-B-TTs (Figure 3C).

The expression levels of VDAC1 were analyzed in tumor protein extract and fixed sections. si-NT-TTs and si-m/hVDAC1-B-TTs were immunoblotted (Figure 3D,E) and IF-stained (Figure 3F,G), showing a decrease of about 55–65% in VDAC1 levels.

The expression levels of the cell proliferation marker KI-67, as analyzed by IHC staining and quantitative analysis, showed a decrease (60%) in the si-m/hVDAC1-B-TTs (Figure 3H,I).

### 3.4. VDAC1 Reduced the Expression of Metabolism-Related Enzymes and Altered the Tumor Microenvironment

The effects of reduced VDAC1 expression levels by si-m/hVDAC1-B treatment of the UM-UC3-derived tumors on tumor metabolism were analyzed following the expression levels of metabolism-related enzymes (Figure 4). The levels of the glucose transporters (GLUT-1) and of the glycolysis enzymes hexokinase (HK-I), glyceraldehyde 3 phosphate (GAPDH) (Figure 4A,B), and of lactate dehydrogenase (LDH) (Figure 4C,D) were highly reduced in si-m/hVDAC1-B-TTs. Similarly, the Krebs cycle enzyme citrate synthase (CS), mitochondrial electron transport complex IVc (COMP IV), and ATP synthase 5a levels were also highly reduced in si-m/hVDAC1-B-TTs (Figure 4C,D).

Next, as cancer cell metabolism is now recognized as modifying the tumor microenvironment (TME) [26], we evaluated the effect of reduced VDAC1 levels on the TME. To follow angiogenesis, we analyzed the expression of the vascular endothelial growth factor (VEGF) family members, VEGF-B, considered to be major mediators of tumor angiogenesis (Figure 5A,C). VDAC1 depletion resulted in a decrease in angiogenesis, as revealed by the immunostaining of VEFG-B relative to their levels in the si-NT-TTs (Figure 5A,C).

Tumor-associated macrophages (TAMs) are one of the most abundant infiltrating immune cells of solid tumors as their accumulation promotes tumor progression. They are present in bladder tumors and play a significant role in BC development [27]. To assess the TAMs, the expression of CD-68, a pan-macrophage marker, also known as macrosialin in mice, was analyzed (Figure 5B,C). The results show that their expression level was highly reduced in si-m/hVDAC1-B-TTs relative to those in si-NT-TTs.

Cancer-associated fibroblasts (CAFs) are the most abundant and critical cellular component of the TME in a variety of solid tumors, producing collagen [28]. Sirius red staining revealed collagen sporadic fibrils found in si-NT-TTs whose levels were highly reduced (Figure 5D,F). Since fibroblasts are the main source of collagen fiber formation, next we IF-stained them for the fibroblast marker alpha smooth muscle actin (α-SMA) using specific antibodies. The α-SMA expression levels in the tumors treated with si-m/hVDAC1-B were also highly reduced (Figure 5E,F), in agreement with the results with Sirius red staining.

Epithelial–mesenchymal transition (EMT) is characterized by a loss of epithelial cell markers, such as cytokeratins and E-cadherin, and the increased expression of mesenchymal cell markers, such as N-cadherin and vimentin. The results show that si-m/hVDAC1-B increased the expression of E-cadherin and decreased the expression levels of N-cadherin and vimentin (Figure 5G–L). The level of N-cadherin decreased, while the expression levels of E-cadherin increased (Figure 5G–J). Vimentin is a type III intermediate filament protein that is expressed in mesenchymal [29]. Similarly, the levels of vimentin in tumors treated with si-m/hVDAC1-B were highly reduced (Figure 5K,L). Low vimentin expression is associated with the inhibition of EMT [30]. These results clearly suggest the inhibition of EMT and, thereby, cell migration.

### 3.5. VDAC1 Depletion Reduced the Expression of Stem Cells

Tumors containing niches enriched for transiently quiescent and self-renewing cells that are essentially cancer stem cells (CSCs) are associated with treatment resistance and prognosis in some cancers. The effect of si-m/hVDAC1-B treatment on CSCs was analyzed by IHC and IF staining for the expression of CSC markers. Cytokeratin 14 (KRT-14), aldehyde dehydrogenase 1a (ALDH1a) [31,32], and the transcription factors SOX2 [33] and CD44 have been shown to initiate bladder cancer [34]. All four CSC markers were expressed in UM-UC3 cell-derived bladder cancer, and their expression was reduced by si-m/hVDAC1-B treatment (Figure 6).

### 3.6. VDAC1 Reduction in Bladder Cancer in the Syngeneic Mouse Model as Induced by the Carcinogen BBN

The second bladder cancer mouse model used in this research was the well-established BBN carcinogen-induced bladder cancer [22]. BBN belongs to the family of nitrosamines, a highly carcinogenic group of compounds commonly found in tobacco smoke. It was administered to male C57/BL6 mice in drinking water (0.05% BBN) for 16 weeks. BBN-induced BC in mice resembles human high-grade invasive urothelial carcinoma, as it recapitulates the same histology and manifests genetic alterations similar to human-muscle-invasive BC [35].

Here, we evaluated the ability of si-m/hVDAC1-B administered intravesically into the bladder by direct injection while visualizing the bladder using ultrasound (UC) apparatus. UC was also used to monitor the bladder and to follow tumor growth. The timeline of cancer development and treatment with si-m/hVDAC1-B (Figure 7A) indicates that by week 12, reactive atypia and dysplasia developed, while between 12 and 20 weeks, carcinoma in situ (CIS) and early muscle invasion were obtained. However, it should be noted that two of the ten BBN-treated mice died during the study either from the injections or being in an advanced stage of BC. The si-NT or si-m/hVDAC1-B used here was encapsulated in nanoparticles made of poly(lactide-*co*-glycolide (PLGA)/polyethylenimine/polyvinyl alcohol PLGA-PEI [16], and the treatment for 4 weeks (twice a week) was started 16 weeks post-BBN treatment to a final concentration of 240 nM, assuming that the bladder volume was 150 μL [36].

Representative UC images of bladders from the control (Figure 7B) and BBN-treated mice, subjected to treatment with si-NT (Figure 7C) or si-m/hVDAC1-B (Figure 7D), showed that the wall layer of the bladder contained tumors occupying the lumen in BBN-treated mice. These were highly reduced in the BBN mice treated with si-m/hVDAC1-B.

At week 20, mice were sacrificed, the bladder was removed and fixed, and the histopathology of the dissected bladder tissues was assessed by hematoxylin and eosin (H&E) staining (Figure 7E–G). Representative images of the H&E staining of bladders from control BBN-untreated mice showed the typical microscopic structure. The bladder wall layers include lining epithelium, the lamina propria, a suburothelial layer composed of an extracellular matrix, capillaries, lymphatics, immune cells, and more and underlie the muscularis propria and the serosa, a thin connective tissue layer that covers the bladder dome (Figure 7E). In BBN-treated mice, the bladder wall structure was damaged and showed a clear tumor occupying the lumen, and also muscle-invasive tumors (Figure 7F). However, in most BBN-treated mice subjected to PLGA-PEI-si-m/hVDAC1-B-treatment, there was a significant decrease in tumor development, and the bladder wall structure appeared to be less damaged, with a more intact epithelium and muscle (Figure 7G).

The destruction of smooth muscle layers is also demonstrated by IF-staining for α-SMA, showing that the intact muscularis propria layer was highly damaged in the bladder from the BBN-treated mice but not in the PLGA-PEI-si-m/hVDAC1-B mice (Figure 8A,B).

With respect to VDAC1 expression levels, high levels were found in the epithelium of bladders from mice and humans (Figure 9A and Appendix A). These high VDAC1 expression levels were also observed in the damaged epithelium of the BBN-treated mice, as well as in the tumors (Figure 9B). In the PLGA-PEI-si-m/hVDAC1-B-treated mice, due to the very high VDAC1 staining intensity, quantitative analysis of the staining was difficult to assess, yet it seems that some decrease was apparent (Figure 9C). This may suggest that PLGA-PEI-si-m/hVDAC1-B given intravesically reached the tumor, but to a lesser extent than to the epithelium.

Similar results with respect to bladder wall layers and tumor development can be seen when staining for VDAC1 expression (Figure 9). In the PLGA-PEI-si-m/hVDAC1-B-treated mice, the bladder smooth muscle layer was more intact (Figure 9C).

It should be noted that there was a variation in the effects of the PLGA-PEI-si-m/hVDAC1-B treatment on the tumor development in the different BBN-treated mice due to differences in the liquid content and composition in the bladder, as well as in the injection position.

## 4. Discussion

In our previous studies, we demonstrated that silencing VDAC1 expression using specific siRNA against human VDAC1 and mouse models of lung, glioblastoma, breast cancer, and others inhibited tumor growth and induced reprogrammed metabolism [14,15,16,17]. This was accompanied by the reversal of most oncogenic properties, resulting in a highly reduced metabolism, angiogenesis inhibition, TME alternation, cancer stem cell elimination, and cell differentiation to normal-like cells [14,15,16,17].

Here, we focused on bladder cancer, testing the effects of VDAC1 silencing using siRNA, recognizing both mouse and human VDAC1 on BC cell lines and two BC mouse models, and demonstrated that reprogramming cancer cell metabolism by VDAC1 silencing resulted in a reversal of oncogenic properties (Figure 10), as found in other cancer types [14,15,16,17].

In addition, for the first time, we chemically induced BC in mice and tested the effect of si-m/hVDAC1-B encapsulated in nanoparticles using intravesical delivery to overcome the challenges of systemic siRNA delivery that are associated with its instability. The results point to si-m/hVDAC1-B as a potential treatment for bladder cancer.

### 4.1. VDAC1 Is Highly Expressed in Bladder Cancer Cell Lines and Tissues

Bladder cancer tissues from human patients show high expression levels of VDAC1 as compared to healthy tissues (Figure 1). Similarly, in UM-UC3 cell-derived BC and BBN-induced BC in mice, the tumors showed high expression levels of VDAC1 relative to its expression in normal mouse bladders. The UM-UC3 BC cell line was found to express the highest VDAC1 level of the tested cancer cell lines. This was due to an increase in VDAC1 per mitochondrion, as the expression levels of the mitochondrial protein CS are similar in the various cell lines. This is similar to what we found for other cancer types [13,14,15,16,17] and is in agreement with the importance of VDAC1 function in regulating energy and metabolite production, as high VDAC1 levels support cancer cell activities.

Interestingly, the bladder epithelium from healthy mice and humans showed very high VDAC1 staining, relative to the muscle and other layers, suggesting a high energy demand and mitochondrial energy reliance.

This is supported by the findings that reduced VDAC1 expression levels in the UM-UC3 and HTB-5 BC cell lines reduced cell viability and mitochondria membrane potential, the driving force for ATP synthesis, accordingly reducing cellular ATP levels (Figure 2). The reduction in the expression levels of cyclin D1 in both of these cell lines suggests an alteration in the cell cycle.

### 4.2. VDAC1 Silencing Reduced the Growth of BC Tumors and Induced Alterations in the Expression of Metabolism-, Microenvironment-, and Cancer-Stem-Cell-Related Proteins

The results presented herein explored metabolism controlled by VDAC1 as an emerging BC treatment target, as demonstrated using two types of BC mouse models. We showed that, as in our previous results obtained with lung, glioblastoma, breast, and other cancer types [13,14,15,16,17,40], VDAC1 depletion resulted in multifactorial responses in cell metabolism, proliferation, angiogenesis, stemness, and differentiation.

VDAC1 silencing in an UM-UC3 cell-derived subcutaneous mouse model resulted in a reduction in the expression of both mitochondrial and glycolytic enzymes. A marked decrease in GLUT-1, HK-I, GAPDH, and LDH, and of Kreb’s cycle and OXPHOS enzymes, supports the likelihood that cancer cells use both glycolysis and OXPHOS [41,42], suggesting that metabolism reprogramming is induced by VDAC1 silencing.

As metabolism is a promising cancer target [43], several drugs targeting specific enzymes such as HK [44] or glutamine metabolism [45] have been developed. However, due to the metabolic plasticity and adaptation of cancer cells, targeting a specific, single protein may not be as effective as modulating cancer cell metabolism by depleting VDAC1, the mitochondria gate keeper.

Cancer cell metabolism is now recognized as modifying the TME [26]. Indeed, cell energy and metabolic reprogramming by VDAC1 depletion leads to alterations in the microenvironment. VDAC1 depletion resulted in a strong decrease in the production of α-SMA, and in collagen and vimentin (Figure 5), suggesting a decrease in the number of CAFs producing them. In bladder cancer, α-SMA- and vimentin-expressing CAFs have been shown to be most prominent in invasive tumors [46]. The reduced expression of VDAC1 also inhibited the infiltration of TMAs, reflected in the decreased expression of the macrophage marker CD-68 (Figure 5B,C).

The reduced expression of VDAC1 also inhibited EMT and cell migration, as it reduced N-cadherin and vimentin and increased E-cadherin expression (Figure 5G–L). Vimentin is also a regulator of cell migration [30].

Increasing evidence supports the hypothesis that tumor CSCs represent a subpopulation of malignant cells that are resistant to conventional cytotoxic/anti-proliferative therapies because they constantly feed the tumor with a supply of cancer cells [37,38,39] and are resistant to some chemotherapeutic strategies [47]. Hence, targeting these CSCs is of utmost importance. In bladder cancer, CSCs are connected to survival and are resistant to conventional treatments such as chemotherapy, radiation, and immunotherapy; thus, targeting these cells in bladder tumors has been proposed as a therapeutic strategy [48].

Here, we analyzed the expression of CSC-related proteins such as cytokeratin-14 (KRT-14), SOX2, CD44, and ALDH1a and demonstrated a decrease in their expression levels upon the administration of si-m/hVDAC1-B treatment (Figure 6). ALDH1a as positive CSCs is found in many cancers such as breast cancer [49], osteosarcoma cancer [50], and BC [51]. Previous studies have reported that KRT-14 and KRT-18 play a pivotal role in regeneration and tumorigenesis [34,52].

CD44 expression levels have been associated with the survival of muscle-invasive BC patients and to their response to chemotherapy and radiotherapy [53]. CD44 levels were correlated with tumor aggressiveness in muscle-invasive BC patients [54], and it was observed that the tumorigenic potential of CD44+ tumor cells in immunocompromised mice was at least 10 to 200 times greater compared to that of CD44 cells [55].

SOX2 expression has been correlated with the presence of CSCs [56] and is upregulated in both mouse and human BC [33] and in BBN-induced BC [33]. SOX2 expression is associated with tumor progression and prognosis [51,57].

Our finding that the expression levels of KRT-14, SOX2, CD44, and ALDH1a decreased upon si-m/hVDAC1-B BC treatment suggests that the metabolic reprogramming of cancer cells via VDAC1 depletion reduces CSC markers, suggesting the inhibition of their proliferation and/or induction of differentiation. Thus, targeting both CSCs and cancer cells by si-m/hVDAC1-B treatment is a promising approach to inhibit BC growth and to offset increased sensitivity to irradiation and chemotherapy. Metabolic flexibility is a hallmark of cancer cell adaptation. Unlike normal stem cells, which use OXPHOS as their primary source of energy, CSCs show metabolic flexibility [58]. In the presence of oxygen, they can switch between OXPHOS and glycolysis to maintain homeostasis, thereby encouraging tumor growth. VDAC1 depletion highly reduced the expression of both glycolysis- and OXPHOS-related enzymes, thus inhibiting both energy and metabolite producing pathways.

### 4.3. si-VDAC1 Delivered Intravesical Attenuated BBN-Driven BC in a Mouse Model

Here, we used the BBN-induced BC [22] mouse model that requires the delivery of si-m/hVDAC1-B to the bladder. Accordingly, we selected intravesical delivery, as is used for BC, and not systemic delivery. In addition, to overcome the challenges of si-m/hVDAC1-B stability, it was encapsulated into PLGA-PEI nanoparticles. As injections of the mice using a catheter were technically impossible, we used ultrasound to visualize the bladder and to monitor the injection location. Moreover, this was the first time that si-VDAC1 encapsulated in PLGA-PEI nanoparticles was injected into the bladder while monitoring it by ultrasound.

As reported in [22], following 16 weeks of BBN treatment, tumors developed in the bladder (Figure 7). BBN-treated mice injected with PLGA-PEI-si-m/hVDAC1-B showed significantly less tumor development. Furthermore, PLGA-PEI-si-m/hVDAC1-B protected against the massive distraction of the smooth muscle layers observed in this group of mice (Figure 8).

It should be noted, however, that due to variations between the different mice, such as the bladder’s liquid content and composition, as observed using the US, in some bladders, si-m/hVDAC1-B encapsulated in PLGA-PEI forms aggregates. Thus, additional studies, including testing other transfection reagents such as Jet-PEI, used here in the s.c. BC mouse model, or other cationic liposomes [59], are required to overcome these technical problems.

Overall, the results of this study using BC cell lines and mouse models point to si-VDAC1 as a potential treatment for bladder cancer.

## Figures and Tables

**Figure 1 cells-13-00627-f001:**
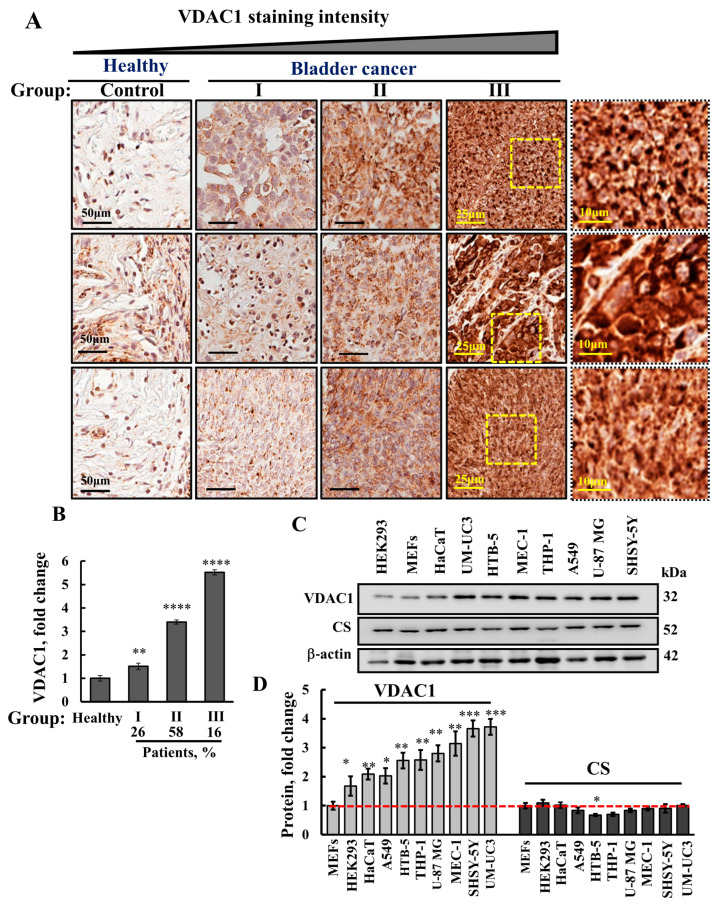
VDAC1 is highly expressed in bladder cancer in human tissues compared to healthy tissues, as well as in bladder cancer cell lines. Immunohistochemical staining of VDAC1 was carried out on human section tissue microarray slides obtained from Biomax US. The array contains bladder sections from both healthy (*n* = 10) and cancerous (*n* = 40) tissues in duplicate cores per case. Selected images from three healthy and nine BC patients and three from each group of low (I), medium (II), and high (III) staining intensity are shown (**A**). Quantitative analysis of the IHC-stained sections of the 20 and 80 samples for healthy and BC tissues, respectively, showing the percentage of patients in each sub-group I, II, and III and indicating the VDAC1 level in each group as the fold of change (**B**). Expression level of VDAC1 and citrate synthase (CS) performed by immunoblot with specific antibodies in different human cancer cell lines and of noncancerous but transformed cells (**C**). Quantitative analysis of VDAC1 and CS levels presented relative to MEF cells (**D**). Results represent the means ± SEM (*n* = 3) * *p* < 0.05, ** *p* < 0.01, *** *p* < 0.001, **** *p* < 0.0001.

**Figure 2 cells-13-00627-f002:**
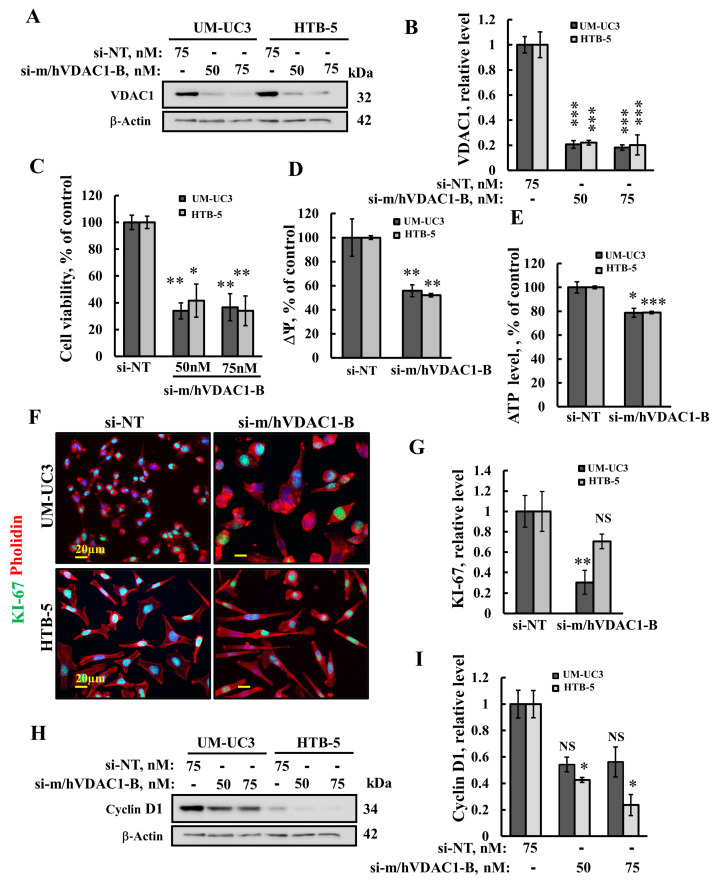
VDAC1 silencing reduces cell viability and proliferation, mitochondrial membrane potential, and ATP levels in bladder cancer cell lines. UM-UC3 and HTB-5 cells were treated with the indicated concentrations of si-m/hVDAC1-B or si-NT, and 96 h post-transfection, cells were assayed for VDAC1 expression level using immunoblotting (**A**) and were quantified (**B**), or for cell viability, assayed with the XTT method (**C**). Cells were analyzed for mitochondrial membrane potential (ΔΨ) (**D**) or cellular ATP levels (**E**) using TMRM (800 nM, 20 min) or luciferase-based assay, respectively, using a plate reader. (**F**,**G**) Cells were treated with, si-m/hVDAC1-B, or si-NT (50 nM), and 72 h post-transfection, were IF-stained for F-actin using phalloidin (red) or for KI-67 (green) using specific antibodies (**F**), and their quantitative analysis is shown (**G**). Immunoblot of cyclin D1 (**H**) and its quantitative analysis (**I**). The results are (*n* = 3) means ± SEM * *p* < 0.05, ** *p* < 0.01, *** *p* < 0.001. NS = non-significant.

**Figure 3 cells-13-00627-f003:**
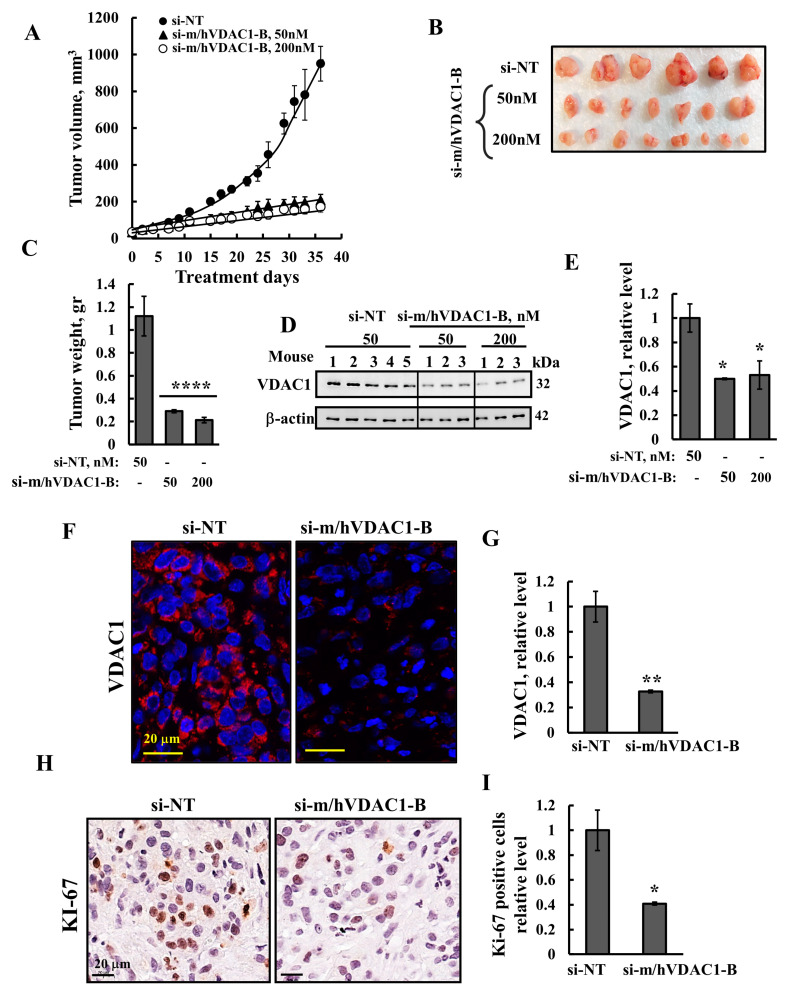
Reduced VDAC1 expression levels inhibited the growth of bladder UM-UC3 cell-derived tumors. (**A**,**B**) UM-UC3 cells were inoculated subcutaneously (SC) into female nude mice (1.2 × 10^6^ cells/mouse in a Matrigel matrix). The tumor sizes were measured (using a digital caliper), and the volume was calculated. When the tumor volume reached ~50 mm^3^, the mice were divided into three groups, and the tumors were injected three times a week intratumorally with si-NT (●, 50 nM) or with si-m/hVDAC1-B (▲, 50 nM; ○, 200 nM). After 35 days, the tumors were dissected, photographed (**B**), and weighed (**C**). (**D**,**E**) si-NT and si-m/hVDAC1-B-TTs (50 nM) were subjected to immunoblotting for VDAC1 (**D**) and its quantification (**E**). (**F**–**I**) Sections of paraffin-embedded si-NT- and si-m/hVDAC1-B-TTs (50 nM) were IF- and IHC-stained for VDAC1 (**F**) and KI-67 (**H**), and staining intensity levels were quantified (**G**,**I**). Results represent the means ± SEM (*n* = 3) * *p* ≤ 0.05; ** *p* < 0.01; **** *p* ≤ 0.0001.

**Figure 4 cells-13-00627-f004:**
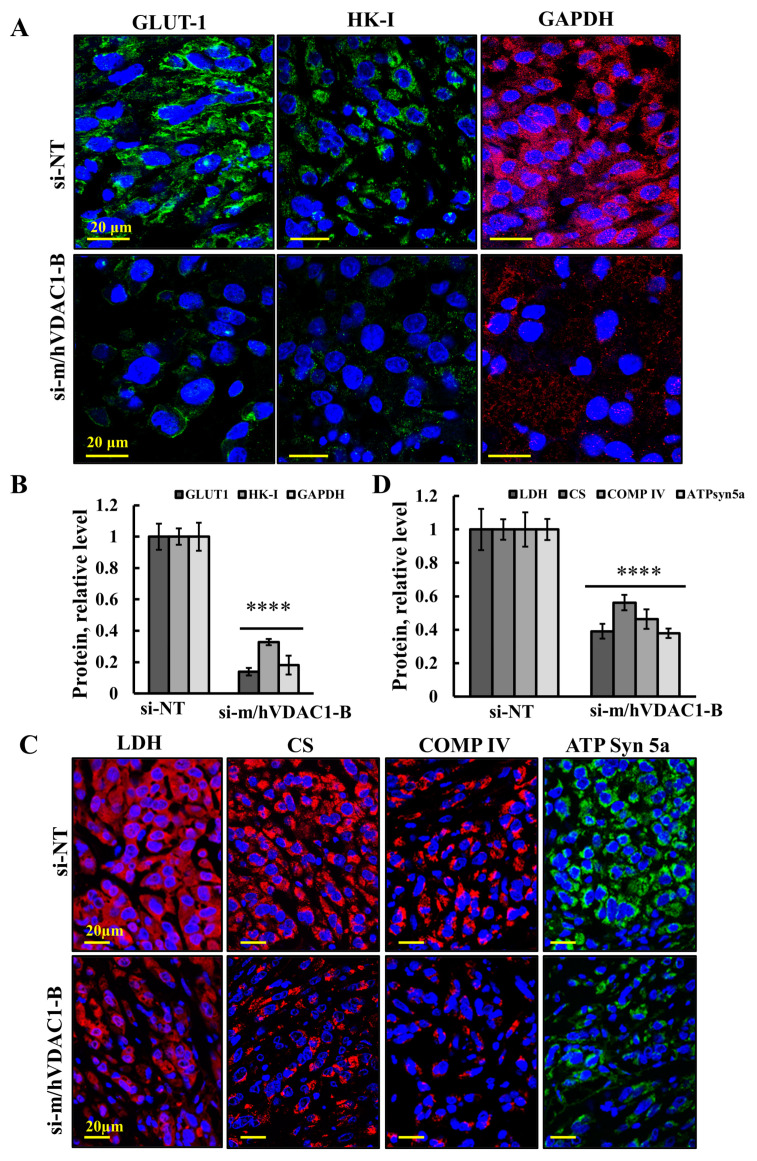
Reduced VDAC1 expression levels silencing decreased the expression of metabolism-related enzymes in UM-UC3 cell-derived tumors. Sections of paraffin-embedded from si-NT and si-m/hVDAC1-B-TTs (50 nM) were IF-stained for HK-I, GLUT-1, and GAPDH (**A**) or LDH, CS, COMP IV, and ATPsyn5a (**C**), and staining intensity levels were quantified (**B**,**D**). Results represent the means ± SEM (*n* = 3). **** *p* < 0.0001.

**Figure 5 cells-13-00627-f005:**
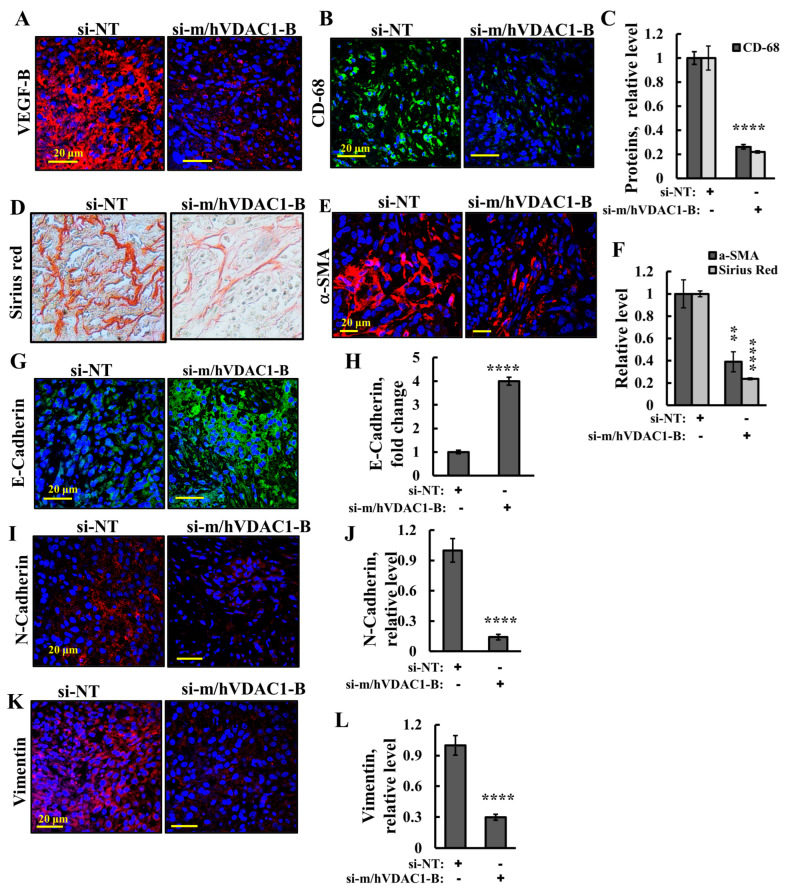
Reduced VDAC1 expression levels altered the tumor microenvironment in UM-UC3 cell-derived tumors. Sections of paraffin-embedded from si-NT and si-m/hVDAC1-B-TTs (200 nM) were subjected to IF staining for VEGF-B or CD-68 (**A**,**B**), and staining intensity levels were quantified (**C**). Sections of paraffin-embedded si-NT- and si-m/hVDAC1-B-TTs (50 nM) were IHC-stained with Sirius red (Magnification 20×) (**D**) or IF-stained for α-SMA (**E**), and staining intensity levels were quantified (**F**). Sections were also stained for E-cadherin and N-cadherin (**G**,**I**) and for vimentin (**K**), with quantified staining intensity levels (**H**,**J**,**L**). Results represent the means ± SEM (*n* = 3). ** *p* < 0.01; **** *p* ≤ 0.0001.

**Figure 6 cells-13-00627-f006:**
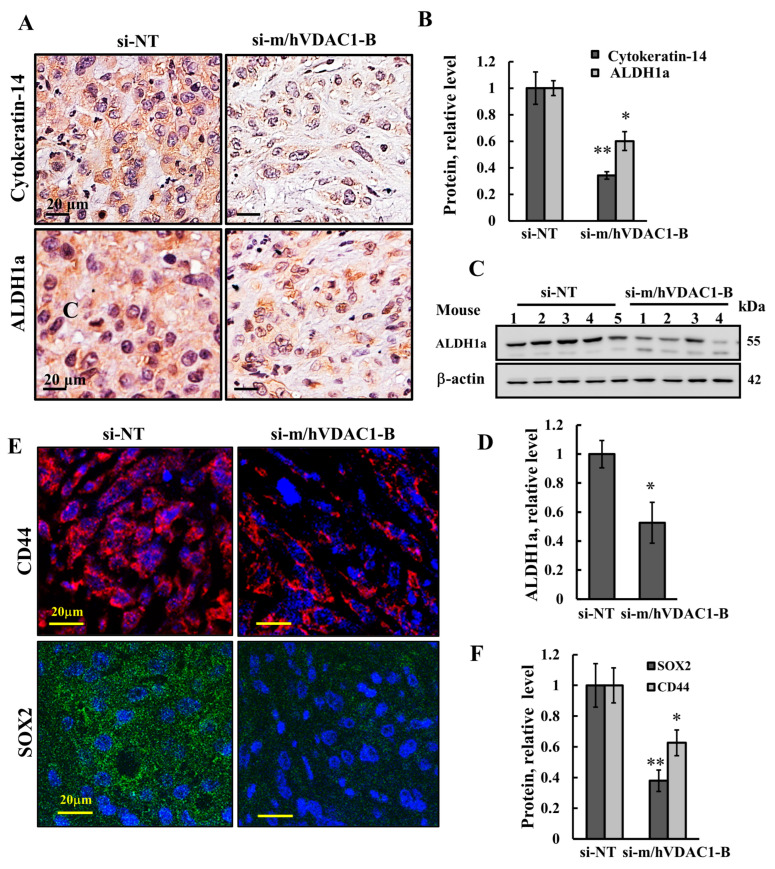
Reduced VDAC1 expression levels decreased the expression of cancer stem cell markers in UM-UC3 cell-derived tumors. Sections of paraffin-embedded si-NT and si-m/hVDAC1-B-TTs (50 nM) were IHC-stained for cytokeratin-14 and ALDH1a (**A**), and staining was quantified (**B**). ALDH1a expression levels were also analyzed by immunoblotting, with β-actin used as a loading control (**C**) and quantified (**D**). IF for CD44 and SOX2 (**E**) and staining intensity levels were quantified (**F**). Results are the means ± SEM (*n* = 4–5), * *p* < 0.05; ** *p* < 0.01.

**Figure 7 cells-13-00627-f007:**
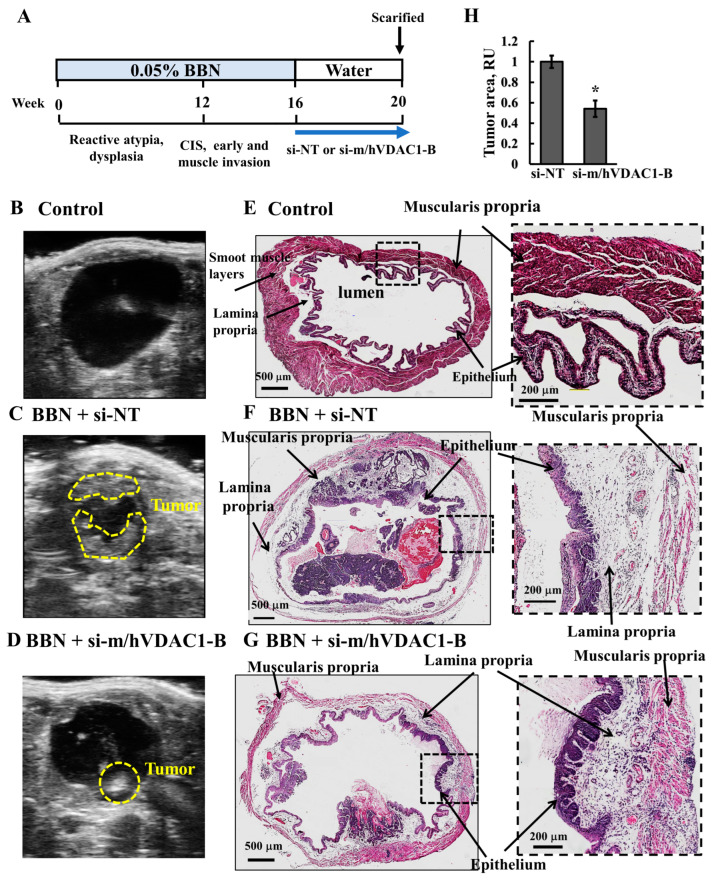
BBN-induced BC and effect of si-m/hVDAC1-B treatment. Schematic presentation of a BBN-induced bladder cancer model, tumor development stages, and the indicated siRNA encapsulated in PLGA-PEI treatment during 16-week treatment with 0.05% BBN given in drinking water (**A**). Representative of US imaging of bladder control (**B**) and BNN-treated mice subjected to si-NT (*n* = 5; 240 nM) (**C**) or si-m/hVDAC1-B (*n* = 3; 240 nM) encapsulated in PLGA-PEI (**D**) treatment. Ultrasound was also used to visualize the bladder and administrate the treatment. Representative images of bladder sections from control and BNN-treated mice subjected to si-NT (*n* = 5; 240 nM) or si-m/hVDAC1-B (*n* = 3; 240 nM) encapsulated in PLGA-PEI treatment stained with hematoxylin and eosin (**E**–**G**). The major structural elements of the bladder are indicated. These include the muscularis (smooth muscle layer), lamina propria, and epithelium. (**H**) Quantitative analysis of the tumor area was calculated using the H&E-stained bladder section, presented as the tumor area relative to the whole bladder area. Results are the means ± SEM (*n* = 3–5), * *p* < 0.05.

**Figure 8 cells-13-00627-f008:**
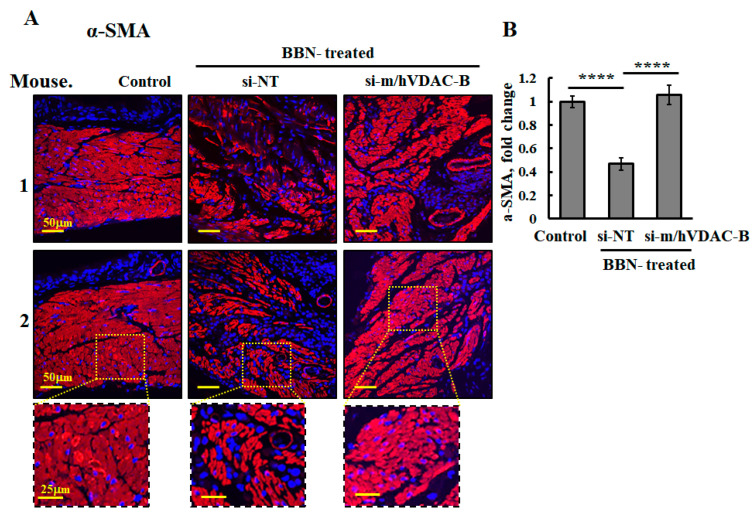
Reduced VDAC1 expression levels prevented BBN-induced destruction of the muscularis smooth muscle layer. Bladder sections obtained from control and BBN-treated mice, treated with si-NT or si-m/hVDAC1-B, were IF-stained with a-SMA to visualize the smooth muscle layer (muscularis) (**A**) and quantify its staining intensity (**B**)—show bladder wall damage as a result of BBN treatment compared to the control, but not when BNN-mice were treated with si-m/hVDAC1-B. **** *p* < 0.0001.

**Figure 9 cells-13-00627-f009:**
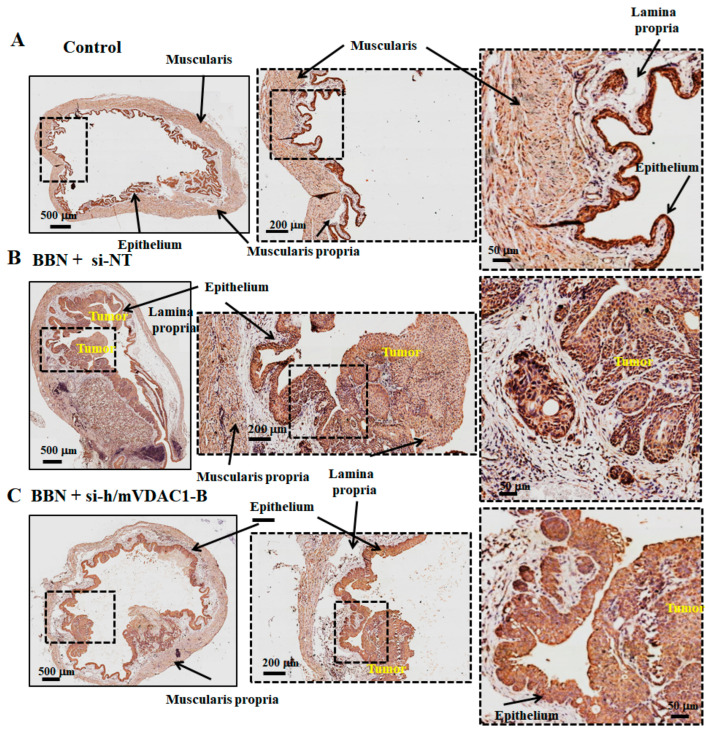
VDAC1 expression in BBN-induced BC with and without si-m/hVDAC1-B treatment. (**A**–**C**) IHC staining for VDAC1 of bladder sections obtained from control and BBN-treated mice, treated with the indicated siRNA at three different magnifications. The major structural elements of the bladder are indicated.

**Figure 10 cells-13-00627-f010:**
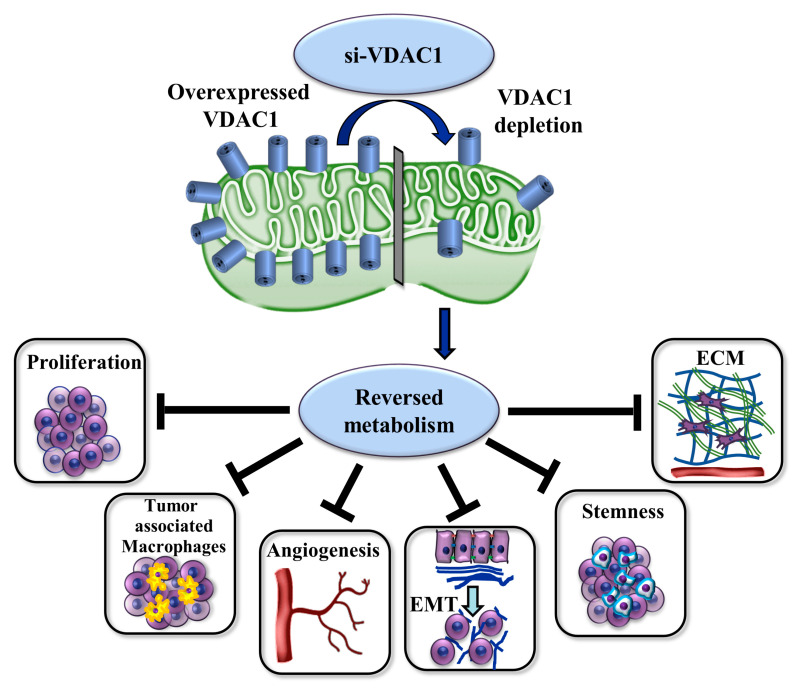
Summary of mitochondrial VDAC1 depletion, leading to metabolic reprogramming and a reversal of oncogenic properties in bladder cancer. In cancer cells, the overexpressed VDAC1 in mitochondria affects homeostatic energy and metabolic state. Silencing VDAC1 expression leads to a reprogramming of metabolism, thereby decreasing energy and the metabolite generation required to support cell growth and survival. Metabolism reprogramming resulted in alterations of the tumor properties, including inhibiting cell proliferation, remodeling of the tumor microenvironment (TME), inhibiting angiogenesis, and decreasing tumor-associated macrophages (TAMs), which can promote tumor growth, invasion, metastasis, and drug resistance. Silencing VDAC1 also inhibited EMT, which is associated with increased cell migration/invasivity and cancer progression. Finally, it reduced the presence of CSCs (stemness) that are resistant to conventional cytotoxic/anti-proliferative therapies, constantly feeding the tumor with a supply of cancer cells [37,38,39].

## Data Availability

All data are presented.

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
