# Peer review of "Silencing the Mitochondrial Gatekeeper VDAC1 as a Potential Treatment for Bladder Cancer"

_cells, 2024, doi:10.3390/cells13070627_

Round 1

Reviewer 1 Report

Comments and Suggestions for Authors

This manuscript has investigated VDAC1 expression in bladder cancers and the effects of VDAC1 downregulation using siRNAs on different aspects of in vitro and in vivo tumour growth. The study augments and supports similar studies using other cancer cell types that have been published previously by these authors.  In general, the study design follows well-established series of experiments characterising the activities of regulators of tumour cell lines and tumour/xenograft growth. The study appears to have been performed well, although further details of some of the methods and results will be required (see below). Figures are generally of a suitable quality, but if the quality of immunohistochemistry images could be improved, this would increase the value of the study and the broader investigation or application of findings in other areas (e.g. pathology). The manuscript is also very well structured and well-written, barring minor grammatical/syntax errors (see below). I feel that the manuscript would be suitable for publication following amendments.

1. There are insufficient and conflicting details on the source and results of immunohistochemical staining of human bladder and bladder tumours. The Methods section contains no details of the source or numbers of human tissues, while in the Results section (3.1), tissues from 3 healthy volunteers and 9 bladder cancer patients is mentioned. In contrast, the Figure 1 legend describes a tissue microarray containing bladder sections from 20 healthy and 80 cancerous tissues. Could the authors please correct and amend the Methods, Results and Figure legend text?

2. As a mitochondrial protein, it is expected that VDAC1 immunostaining would be cytoplasmic, however in a number of the images, nuclear immunoreactivity seems to be quite prominent. This is only evident in immunohistochemically stained sections where immunoreactivity is detected using DAB and not immunofluorescence microscopy. What do the authors feel is the cause of this? The quality (clarity, magnification) of immunohistochemically stained sections could be improved.

3. When bladder cancer cell line xenografts were treated with intratumoral injections of siRNAs, was downregulation of VDAC1 expression, as depicted in Figure 3H, only evident surrounding injection sites?

4. Further details of the BBN-induced bladder cancer model and results should be added to the manuscript. The numbers of mice used for each treatment group should be included in the Methods section, and because numbers overall will not be high, figures depicting tumour burden (as estimated by ultrasound) vs time, could be added (how often was ultrasound performed to monitor tumour growth?). Were the mice given a single dose of the si-VDAC1/control treatments at 16 weeks? This should be specifically stated. At present, it is not clear how the authors were calculating or estimating tumour burden, and in addition, the authors state that ‘several’ mice died during the experiments, and it is essential that such details are also precisely listed.

5. The authors state “In the PLGA-PEI-si-m/hVDAC1-B-treated mice, VDAC1 staining was slightly, but significantly decreased” (lines 495-496). However, reduced VDAC1 staining intensity is not clearly evident in the images included in Figure 9. Inclusion of the term ‘significantly’ implies use of a statistical test, details of which are not included in the manuscript. Could the authors provide further information, including methods (how staining was assessed) and evidence (e.g. images, calculations / results of statistical tests) to support their conclusions?

6. The authors frequently use the term ‘silencing’ of VDAC1, however their own results, particularly in tumours, clearly show that VDAC1 remains highly expressed. It is suggested that an alternative term is used as this becomes confusing when describing and interpreting results or mechanisms of action of siRNA treatments.

Comments on the Quality of English Language

There are minor English language errors that require correction (e.g. “destructed” (sic), ‘distraction’ instead of ‘destruction’, etc). Otherwise, English language (grammar) is very good.

Author Response

  1. There are insufficient and conflicting details on the source and results of immunohistochemical staining of human bladder and bladder tumours. The Methods section contains no details of the source or numbers of human tissues, while in the Results section (3.1), tissues from 3 healthy volunteers and 9 bladder cancer patients is mentioned. In contrast, the Figure 1 legend describes a tissue microarray containing bladder sections from 20 healthy and 80 cancerous tissues. Could the authors please correct and amend the Methods, Results and Figure legend text?

 Ans - We have now provided details about the tissue array obtained from Biomax in the Methods section (lines 250-252), the results (lines 271–­276), and in the Figure 1 legend (lines 443–447).  In Figure 1A, representative images from three healthy and nine BC patients are presented. The IHC quantification was carried out on all samples—a total of 10 and 40 samples in duplicate cores per case for healthy and BC tissues, respectively.

  1. As a mitochondrial protein, it is expected that VDAC1 immunostaining would be cytoplasmic, however in a number of the images, nuclear immunoreactivity seems to be quite prominent. This is only evident in immunohistochemically stained sections where immunoreactivity is detected using DAB and not immunofluorescence microscopy. What do the authors feel is the cause of this? The quality (clarity, magnification) of immunohistochemically stained sections could be improved..

Ans -The imaging of IHC slides were carried out using the panoramic scanner. Because this microscope captures a thick layer of the section, the cytosol with the mitochondria could be overlapped with the nucleus.  IF imaging using confocal microscopy allowed us to focus on different layers and thin sections; therefore, we did not observe VDAC1 staining in the nucleus.

  1. When bladder cancer cell line xenografts were treated with intratumoral injections of siRNAs, was downregulation of VDAC1 expression, as depicted in Figure 3H, only evident surrounding injection sites?

Ans - Figure 3H represents nucleus staining with Ki-67.  Yet to address your question, we have now added the injection pattern to the Method section (lines 207–211). Tumors intratumorally treated with the injection volume were 10–15% of the tumor volume.  To ensure that the siRNAs could reach all of the tumor volume, the tumors were injected at a single point (bolus) if the tumor was small, and at up to three different boluses for a large tumor.  In addition, the tumor injection points were not constant, and were rotated every injection

It should be noted that images were taken from different fields of the tumor section to represent the whole tumor area.

This information is now added to the Methods section (lines 247–248).

  1. Further details of the BBN-induced bladder cancer model and results should be added to the manuscript. The numbers of mice used for each treatment group should be included in the Methods section, and because numbers overall will not be high, figures depicting tumour burden (as estimated by ultrasound) vs time, could be added (how often was ultrasound performed to monitor tumour growth?). Were the mice given a single dose of the si-VDAC1/control treatments at 16 weeks? This should be specifically stated. At present, it is not clear how the authors were calculating or estimating tumour burden, and in addition, the authors state that ‘several’ mice died during the experiments, and it is essential that such details are also precisely listed.

Ans - The missing information is now added to the Methods section (lines 207–211, 218-220), to the legend of Figure 7, and also to lines 636-637.

  1. The authors state “In the PLGA-PEI-si-m/hVDAC1-B-treated mice, VDAC1 staining was slightly, but significantly decreased” (lines 495-496). However, reduced VDAC1 staining intensity is not clearly evident in the images included in Figure 9. Inclusion of the term ‘significantly’ implies use of a statistical test, details of which are not included in the manuscript. Could the authors provide further information, including methods (how staining was assessed) and evidence (e.g. images, calculations / results of statistical tests) to support their conclusions?

Ans -We agree with this reviewer’s comment that the use of the word significant should be based on quantification. We have now modified this to read:  In the PLGA-PEI-si-m/hVDAC1-B-treated mice, due to the very high VDAC1 staining intensity, quantitative analysis of the staining was difficult to assess, yet it seems that some decrease was apparent (Figure 9C).

  1. The authors frequently use the term ‘silencing’ of VDAC1, however their own results, particularly in tumours, clearly show that VDAC1 remains highly expressed. It is suggested that an alternative term is used as this becomes confusing when describing and interpreting results or mechanisms of action of siRNA treatments.

Ans -Indeed, our siRNA treatment at the experimental setting of the intratumor injection and the frequency of treatment used, VDAC1 levels were decreased by about 60% (Fig. 3F). However, because the mechanism of reducing VDAC1 expression involves silencing gene expression, this word refers to the mechanism of reducing expression.  However, we replaced the word “silencing” with “reduced expression” when appropriate throughput the manuscript.

Comments on the Quality of English Language

There are minor English language errors that require correction (e.g. “destructed” (sic), ‘distraction’ instead of ‘destruction’, etc). Otherwise, English language (grammar) is very good.

Ans - Although the paper was edited by an expert, we have subjected it to second English editing.

Reviewer 2 Report

Comments and Suggestions for Authors

The manuscript explores the potential of siRNA-based therapy targeting VDAC1 for treating bladder cancers (BC). The manuscript's strengths lie in its precise method descriptions, facilitating experiment replication, and the effective use of mouse models for research. The results compellingly underscore the role of VDAC1 in BC.

MAJOR POINT

1.     Section 2.3. siRNA Transfection: The design of VDAC1-specific siRNA involves certain nucleotide sequences, which, although relatively short, may inadvertently target other mRNAs. BLAST software analysis reveals potential complementarity to other genes, such as Homo sapiens BAC and peroxisomal biogenesis factor 3 (PEX3).

Details of analysis: AC231643.3 and NG_008459.1, for BAC and PEX3 respectively.  BLAST was performed toward the sense silencing sequence.

It would be beneficial for the authors to discuss the possibility of silencing unintended targets using the same silencing sequences.

MINOR POINTS

  1. Section 2.3. siRNA Transfection: The authors should specify the type of transfectant used for silencing (JeT Prime transfection reagent), whether it is lipofectamine-based, PEI-based, or another type.
  2. The cell lines UM-UC3 and HTB-5 from Homo sapiens were used for target silencing (section 3.2). However, the abstract suggests that the designed VDAC1-specific siRNA could silence both mouse and human VDAC1. Are there experimental proofs of VDAC1 silencing in mouse-derived cell lines?
    1. In the acknowledgments section, two individuals are listed. It would promote transparency if their declaration of not wanting to be listed as authors, if their contribution was substantial, were included. While unconventional, such transparency enhances credibility.
    2. The funding is unclear, with only "This research was supported by Krieger Foundation" provided. Should we request further details regarding the extent of the funding?

Author Response

 Section 2.3. siRNA Transfection: The design of VDAC1-specific siRNA involves certain nucleotide sequences, which, although relatively short, may inadvertently target other mRNAs. BLAST software analysis reveals potential complementarity to other genes, such as Homo sapiens BAC and peroxisomal biogenesis factor 3 (PEX3).

Details of analysis: AC231643.3 and NG_008459.1, for BAC and PEX3 respectively.  BLAST was performed toward the sense silencing sequence.

It would be beneficial for the authors to discuss the possibility of silencing unintended targets using the same silencing sequences.

Ans- We appreciate this comment, but will not discuss this possible off-target effect in the current paper. Indeed, performing BLAST on the sense sequence, we found high homology with NG_008459.1, PEX3, but not with BAC. We plan to check this siRNA effect on the expression of PEX3. If we find that its expression is reduced, we will test siRNA specific to this protein on cell proliferation and cell viability.  It should be noted that we found publication for RNAi-mediated silencing of PEX6, but not for PEX3 silencing.

MINOR POINTS

  1. Section 2.3. siRNA Transfection: The authors should specify the type of transfectant used for silencing (JeT Prime transfection reagent), whether it is lipofectamine-based, PEI-based, or another type.

Ans- jetPRIME is a cationic polymer-based, polyvalent transfection reagent suited for DNA and siRNA delivery. We added to the Materials section that it is a polyvalent transfection reagent (line 111).

2. The cell lines UM-UC3 and HTB-5 from Homo sapiens were used for target silencing (section 3.2). However, the abstract suggests that the designed VDAC1-specific siRNA could silence both mouse and human VDAC1. Are there experimental proofs of VDAC1 silencing in mouse-derived cell lines?

Ans- We published the activity of the si-m/h-VDAC1-B in both mouse and human cells (Pandey S, Machlof-Cohen R., Santhanam, Shteinfer-Kuzmine A, Shoshan-Barmatz, V. (2022) Silencing VDAC1 to Treat Mesothelioma Cancer: Tumor Reprograming and Altering Tumor Hallmarks, Biomolecules Jun 27;12(7):895.  It is now added, line 291.

The figure is in the attached PDF .

Round 2

Reviewer 1 Report

Comments and Suggestions for Authors

The authors have address some of the reviewers’ comments, however there are several areas that require clarification.

1. Immunohistochemical staining of human bladder and bladder tumours:- The authors have clarified the source and number of specimens included in the tissue microarray used for this study (lines 249-251), which contains duplicate samples from 10 healthy and 40 bladder cancer patients.

The purpose of including duplicate cores in tissue microarrays is to provide some indication of variability of expression of target proteins being investigated, to account for variability of cell types (tumour, stroma, vessels) within tumour tissues, or to account for loss of cores during processing.

Duplicate samples from an individual tumour are not separate samples unless they were specifically chosen to be (for example, necrotic vs non-necrotic areas of a tumour). Investigators could either use the average (mean) of results obtained from each of the cores from the same tumour, or for some studies, selection of the highest or lowest score for each patient would be appropriate.

In the Figure 1 legend, it seems that the authors are treating each of the tumour cores as if it were derived from a separate patient and there does not appear to be any justification for doing so. However, in Figure 1B, the X-axis label states “Patients”. Please clarify how results for each patient were derived. As loss of tissue cores is common when using microarrays, it would be appropriate to simply state in the Results text “VDAC1 immunostaining was able to be scored in _ cores (_ patients).” The statement in the Figure 1 legend (“thus, there are a total of 20 and 80 samples for health and BC tissues, respectively”) can be removed as the authors have just stated that duplicate cores for 10 healthy and 40 cancerous tissues were included.

2. Scoring of immunohistochemically stained sections: How were the 1.5-, 3.5- and 5.5-fold chosen to describe the scoring? It seems as though scoring results should be a continuous variable, however the graph in Figure 1B indicates that scoring was very tightly grouped around these values. Was scoring only of cytoplasmic VDAC1 staining? This type of information can be simply stated in the Methods section (there is no need to repeat details of methods in the results section). Because VDAC1 staining is novel, it would be usual to depict images illustrating each of the scoring groups. If this is what the authors were intending to show in Figure 1A, specific details should be added to the figure legend. The size of the magnification bars in Figure 1A should also be added to the figure legend. (Labelling of magnification bars is also missing in other figures).

3. Inclusion of additional details of in vivo models has clarified methods and results interpretation. It appears that for the BBN-induced tumour model, results for si-m/hVDAC1-B are based on only 3 mice, which is not sufficient to obtain definitive findings, but this is alluded to in the Discussion section. It is interesting that additional details include that advanced BC was a cause of death in this group, indicating that si-m/hVDAC1-B treatment was not effective for all tumour-bearing mice (lines 376-377). This is not unexpected for chemically-induced tumours, which will have more diverse genotypes (and phenotypes) compared to simpler tumour models that are based on growth of cell lines as xenografts. However, the inadequate cell numbers mean that at this stage it cannot be deduced whether >25% of tumours (3 of 4 or 5) may be resistant to si-m/hVDAC1-B treatment, or whether resistance to the treatment is higher or lower. Reference to the susceptibility/resistance of BBN-induced tumours to si-m/hVDAC1-B treatment could be added to the manuscript.

Comments on the Quality of English Language

English language editing is patchy. The word “destructed” is not English and should be replaced.

Author Response

We thank this reviewer for the additional comments that we have addressed as presented below and when appropriate introduced to the revised version (in blue).

The authors have addressed some of the reviewers’ comments, there are several areas that require clarification.

  1. Immunohistochemical staining of human bladder and bladder tumours: The authors have clarified the source and number of specimens included in the tissue microarray used for this study (lines 249–251), which contains duplicate samples from 10 healthy and 40 bladder cancer patients.

The purpose of including duplicate cores in tissue microarrays is to provide some indication of variability of expression of target proteins being investigated to account for variability of cell types (tumour, stroma, vessels) within tumour tissues, or to account for loss of cores during processing.

Duplicate samples from an individual tumour are not separate samples unless they were specifically chosen to be (for example, necrotic vs. non-necrotic areas of a tumour). Investigators could either use the average (mean) of results obtained from each of the cores from the same, or for some studies, selection of the highest or lowest score for each patient would be appropriate.

We thank this reviewer for the detailed explanation for the meaning of including duplicates in the tissue array. The stained section from each patient sample was homogeneous, and we did note a significant difference in staining intensity in the stained section.

In the Figure 1 legend, it seems that the authors are treating each of the tumour cores as if it were derived from a separate patient and there does not appear to be any justification for doing so. However, in Figure 1B, the X-axis label states “Patients”. Please clarify how results for each patient were derived. As loss of tissue cores is common when using microarrays, it would be appropriate to simply state in the Results text “VDAC1 immunostaining was able to be scored in _ cores (_ patients).” The statement in the Figure 1 legend (“thus, there are a total of 20 and 80 samples for health and BC tissues, respectively”) can be removed as the authors have just stated that duplicate cores for 10 healthy and 40 cancerous tissues were included.

We tried to explain the quantitative analysis of the staining intensity. As suggested, the sentence "thus, there are a total of 20 and 80 samples for health and BC tissues, respectively" is now removed.

  1. Scoring of immunohistochemically stained sections: How were the 1.5-, 3.5- and 5.5-fold chosen to describe the scoring? It seems as though scoring results should be a continuous variable, however the graph in Figure 1B indicates that scoring was very tightly grouped around these values. Was scoring only of cytoplasmic VDAC1 staining? This type of information can be simply stated in the Methods section (there is no need to repeat details of methods in the results section). Because VDAC1 staining is novel, it would be usual to depict images illustrating each of the scoring groups. If this is what the authors were intending to show in Figure 1A, specific details should be added to the figure legend. The size of the magnification bars in Figure 1A should also be added to the figure legend. (Labelling of magnification bars is also missing in other figures).

We observed differences in the staining intensity between different patient samples. Using the patients’ available data, we could not find a clear link between the stage of the disease and the staining intensity. Therefore, we defined three group intensities: low, medium, and high, and their analysis resulted in increased VDAC1 expression by 1.5-fold, 3.5-fold, and 5.5-fold.

As suggested, we now added to Fig. 1A, group I, II, and IIl, and use this also in Fig. 1B. We added the following to the figure legend: Selected three images from healthy and nine BC patients that were sub-grouped to low (I), medium (II, and high (III) staining intensity are shown (A). Quantitative analysis of the IHC-stained sections of the 20 and 80 samples for healthy and BC tissues, respectively, showing the percentage of patients in each sub-group I, II, and III and indicating the VDAC1 levels in each group as the fold of change.

This is also indicated in the Results section (lines 273-274) and the current figure is now replaced with the modified figure.

As to: Was scoring only of cytoplasmic VDAC1 staining?

The scoring is of the staining intensity of the whole section, as performed by the Panormic microscope.

The missing magnification bars have now been added.

  1. Inclusion of additional details of in vivo models has clarified methods and results interpretation. It appears that for the BBN-induced tumour model, results for si-m/hVDAC1-B are based on only 3 mice, which is not sufficient to obtain definitive findings, but this is alluded to in the Discussion section. It is interesting that additional details include that advanced BC was a cause of death in this group, indicating that si-m/hVDAC1-B treatment was not effective for all tumour-bearing mice (lines 376-377). This is not unexpected for chemically-induced tumours, which will have more diverse genotypes (and phenotypes) compared to simpler tumour models that are based on growth of cell lines as xenografts. However, the inadequate cell numbers mean that at this stage it cannot be deduced whether >25% of tumours (3 of 4 or 5) may be resistant to si-m/hVDAC1-B treatment, or whether resistance to the treatment is higher or lower. Reference to the susceptibility/resistance of BBN-induced tumours to si-m/hVDAC1-B treatment could be added to the manuscript.

We have pointed out several factors that may affect the degree of response (lines 408–411): It should be noted that there was a variation in the effects of the PLGA-PEI-si-m/hVDAC1-B treatment on the tumor development in the different BBN-treated mice due to differences in the liquid content and composition in the bladder, as well as in the injection position.

We did not consider resistance to the siRNA as the treatment was for a relatively short time (4 weeks).  Indeed, many RNA viruses escape RNAi-mediated suppression by counteracting the RNAi machinery through mutation of the targeted region by encoding viral suppressors or both (Development of resistance to RNAi in mammalian cells Ann N Y Acad Sci. 2005 Nov; 1058: 105–118, PMID: 16394130). However, as indicated in this review, development of RNAi resistance in mammalian cells could be because of any possible blocks in RNAi pathways.

Although we cannot rule out development of resistance to siRNA, as currently this is an issue that is not considered or discussed in studies using siRNA, we believe we should not discuss this as one of the possible reasons for the variation in the response degree (presented above).

Comments on the Quality of English Language

English language editing is patchy. The word “destructed” is not English and should be replaced.

Sorry, in the revised version we have replaced this word with “damaged, and missed it in 3 places, now these were also replaced.
